# An Efficient Compression Method for Lightning Electromagnetic Pulse Signal Based on Convolutional Neural Network and Autoencoder

**DOI:** 10.3390/s23083908

**Published:** 2023-04-12

**Authors:** Jinhua Guo, Jiaquan Wang, Fang Xiao, Xiao Zhou, Yongsheng Liu, Qiming Ma

**Affiliations:** 1School of Mathematics and Physics, Shanghai University of Electric Power, Shanghai 201306, China; 2Institute of Electrical Engineering, Chinese Academy of Sciences, Beijing 100190, China; 3Institute of Solar Energy, Shanghai University of Electric Power, Shanghai 200090, China

**Keywords:** lightning, deep learning, feature compression, autoencoder, convolutional neural network

## Abstract

Advances in technology have facilitated the development of lightning research and data processing. The electromagnetic pulse signals emitted by lightning (LEMP) can be collected by very low frequency (VLF)/low frequency (LF) instruments in real time. The storage and transmission of the obtained data is a crucial link, and a good compression method can improve the efficiency of this process. In this paper, a lightning convolutional stack autoencoder (LCSAE) model for compressing LEMP data was designed, which converts the data into low-dimensional feature vectors through the encoder part and reconstructs the waveform through the decoder part. Finally, we investigated the compression performance of the LCSAE model for LEMP waveform data under different compression ratios. The results show that the compression performance is positively correlated with the minimum feature of the neural network extraction model. When the compressed minimum feature is 64, the average coefficient of determination $R^{2}$ of the reconstructed waveform and the original waveform can reach 96.7%. It can effectively solve the problem regarding the compression of LEMP signals collected by the lightning sensor and improve the efficiency of remote data transmission.

## 1. Introduction

Existing real-time VLF/LF lightning signal acquisition equipment needs to acquire a large number of lightning signals, especially in the event of thunderstorms. The compressed transmission of the lightning signal is a critical part of the process. A good compressed transmission method not only reduces time costs, but also facilitates faster identification and localization of lightning events [1].

With the development of machine learning, more and more fields have begun to seek research methods combined with it. In the direction of deep learning [2], thanks to the powerful data representation ability of its model method, it has been widely used in biomedicine, image recognition, face detection and emotion classification fields [3,4,5,6] in recent years. Simultaneously, machine learning has also made some progress in the study of lightning. In the prediction study of lightning occurrence, Mostajabi et al. [7] used the Automated Machine Learning (AutoML) method to select XGBoost, a machine learning algorithm, to predict the occurrence of imminent lightning. Kamangir et al. [8] developed a neural network; this model has predicted the occurrence of thunderstorms in a 400 km^2^ area in southern Texas. In the recognition and classification task of lightning data, Morales et al. [9] compared Multi-resolution analysis (MRA) with Artificial Neural Network (ANN), K-Nearest Neighbors (KNN) and Support Vector Machine (SVM)—which are machine learning methods—to analyze transmission line lightning events and achieve good classification results. Zhu et al. [10] used the SVM algorithm to classify representative cloud-to-ground and intracloud lightning, and its accuracy could reach 97%. In locating lightning events, Karami et al. [11] proposed using a machine learning method to locate the lightning strike point based on the lightning-induced voltage value measured by sensors on the transmission line. Recently, Wang et al. [12] also combined the lightning positioning method with artificial intelligence, and used the deep-learning encoding feature matching method to improve the speed, accuracy and anti-interference ability of the original positioning algorithm.

In addition, the method of machine learning is also applied to the estimation of lightning strike probability [13], the prediction of natural disasters caused by lightning [14], and the detection of lightning patterns [15]. In the past few years, the addition of artificial intelligence techniques has facilitated the study of lightning events, and as scientific and technological advances have led to more accurate collection of lightning data by lightning sensor networks, the amount of data has increased, and a new method for the extraction and compression of lightning electromagnetic pulse features has become a major issue. This is similar to artificial-intelligence-based methods for the compressed sensing of ECG signals [16,17], and feature the extraction of hyperspectral signals [18]. For example, compression and reconstruction of ECG signals using compressed sensing methods can achieve noise reduction processing of ECG signals while taking into account signal integrity [19].

This study establishes a new artificial intelligence model called LCSAE, which uses a stacked autoencoder (SAE) and a convolutional neural network to compress the features of LEMP waveform data. The original LEMP dataset consists of 1000 features, and the LCSAE model can compress these features by adjusting the compression ratio. Experimental results demonstrate a positive correlation between the LEMP waveform compression performance and the minimum compression features used in LCSAE. When the minimum compression feature is set to 64, the $R^{2}$ between the reconstructed and original waveforms reaches 96.7%. These results indicate that the model effectively achieves the target of compressing the LEMP signal.

The specific structure of the article is as follows: Section 2 presents the LEMP dataset and lightning sensor devices used in our experiments, along with the details and specific structure of our novel LCSAE neural network model designed for dimensionality reduction and compression. Section 3 gives the experimental results and evaluates the feature compression effect of the proposed neural network model. Section 4 concludes and discusses the whole paper.

## 2. Model and Methods

Based on the deep learning stacked autoencoder model and the collected LEMP waveform data of the real-time three-dimensional (3D) lightning detection network, we proposed an LCSAE waveform compression and reconstruction method. This method can compress the collected lightning data with a length of 1000 features according to a fixed ratio, while the original waveform can be reconstructed using the compressed feature code.

Figure 1 is a schematic diagram of the overall process of the proposed waveform compression method. The sensor on the left side of the figure acquires LEMP waveform data, and the acquired waveform data set is transmitted to the LCSAE input layer on the right side. The waveform data is compressed by the encoder to obtain a low-dimensional feature code, and this code can be decoded by a decoder structure to obtain a reconstructed waveform. Ideally, the reconstructed waveform output by the output layer should be consistent with the original waveform.

The above LCSAE method is an unsupervised deep learning model based on an autoencoder. The following will start from the model method and the debugging selection of the main structure in the model building process.

### 2.1. Data Sets Collected by Sensor Networks

The LEMP waveform data used in this study were collected by the 3D Lightning Detection Network of the Institute of Electrical Engineering of the Chinese Academy of Sciences [20], which covers the Asia-Pacific region and has more than 400 sensor detection sites so far. The sampling rate of each site is 1 MSPS, the length of a single sampling time is 1 ms, and the characteristic length of each collected lightning electromagnetic pulse waveform data is 1 ms. Figure 2 shows LEMP waveform data collected from a sensor device, which consists of 1000 feature points. Figure 3 depicts a block diagram of the lightning sensor device, comprising of antennas, a field-programmable gate array (FPGA), an advanced RISC machine (ARM), and an internet and storage module. The magnetic antenna detects and outputs the magnetic field signal, while the FPGA and ARM process the digital signal. The collected lightning signals can be stored and transmitted through the internet.

The experiment used 30,000 LEMP data in the database, which cover six types of important LEMP waveform data, each with 5000 signal samples, namely negative cloud-to-ground flash (−CG), positive cloud-to-ground flash (+CG), negative narrow bipolar event (−NBE), positive narrow bipolar event (+NBE), cloud ground flash with ionosphere-reflected signals (CG-IR) and far-field skywave (SW), where −NBE, +NBE and CG-IR are cloud flashes. Figure 4 is an example of one of the six types of data waveforms. Before using the deep learning model for training, the data set used is shuffled, 10% of the sample data is taken as the test set, 10% of the remaining data is taken as the validation set, and 90% is used as the training set.

### 2.2. Autoencoder

The autoencoder was first developed in the 1980s, and this method can convert high-dimensional data into low-dimensional codes [21]. As early as 1986, Rumelhart et al. [22] proposed the concept of an autoencoder, and then in 1988, Bourlard et al. [23] gave a detailed explanation of the autoencoder. In the following years, the sparse autoencoder, denoising autoencoder, puncture autoencoder and variational autoencoder [24,25,26,27] have been successively proposed. With the advantages of simple training processes, easy multi-layer stacking, and a good generalization performance, autoencoders and their improved algorithms have been successfully applied in research work such as anomaly detection, machine translation, and scene segmentation [28,29,30]. The widely used transformer model [31] is based on the autoencoder. This paper draws inspiration from this and builds a new deep learning model: Lightning-Stacked Autoencoder (LCSAE), which is used for LEMP data compression work.

A simple autoencoder model consists of only an input layer, an output layer, and a hidden layer, as illustrated in Figure 5. The principle is to minimize the error between the input *x* and the output $\hat{x}$ by learning an effective encoding of the intermediate hidden layer *h* [21], which can be achieved through dimensionality reduction if we set a smaller feature value for *h*.

The autoencoder is divided into two parts: the encoder and the decoder. The main task of the encoder is to perform feature compression on the input data and learn low-dimensional representations of high-dimensional data. The decoder reconstructs the signal data according to the features compressed by the encoder. This process can be further illustrated by Equation (1):(1)$$\hat{x}=g\left(y\right)=g\left(f\left(x\right)\right)$$
where x represents the input signal, $f\left(\cdot \right)$ refers to the encoding process of the encoder, which is the encoded signal after the encoder, $g\left(\cdot \right)$ represents the decoding process of the decoder, and $\hat{x}$ is the output signal. The learning objective of the autoencoder is to minimize the reconstruction error, ideally $\hat{x}=x$.

### 2.3. Structure of Compression and Reconstruction Model LCSAE

SAE is obtained by adding multiple hidden layers on the basis of the autoencoder. Compared to traditional autoencoders, SAE has a stronger data representation ability. The LCSAE model is based on an SAE in which a convolutional neural network approach is added to enhance the representation and extraction of lightning occurrence signals. The convolutional neural network is a popular network structure in deep learning. This kind of network contains structures such as convolutional layers, pooling layers, and fully connected layers [32]. The convolutional layer can process data of different dimensions, such as waveform signals and image data, of which 1D-Convolutional (one-dimensional convolutional) is often used in signal processing, such as in the processing of ECG signals [33,34]. Both the ECG signal and the LEMP signal have similar waveform structure characteristics. The LCSAE model we built uses 1D-Convolutional to extract features from LEMP data. The transfer between convolutional layers is as follows (2):(2)$$x_{l+1}=f\left(w*x_{l}+b\right)$$
where $x_{l}$ and $x_{l+1}$ are the feature vectors of the input signal and output signal, respectively. The symbol ‘∗’ is the convolution operator, w is the weight matrix between layers, b is the bias, and $f\left(\cdot \right)$ is the activation function between the transfer layers.

The pooling layer is a dimensionality reduction method in neural networks. Common ones include max pooling and average pooling. We use max pooling (MaxPooling) to reduce the dimensionality of the input features of the previous layer. The calculation is published as follows (3)
(3)$$N_{l+1}=MAX\left(\frac{N_{l}-p+1}{r}\right)$$
where $N_{l}$ and $N_{l+1}$ represent the number of neurons at layer l and layer l+1, respectively, p is the pooling size, and r is the step size.

The LCSAE model designed in the experiment consists of three parts: Encoder, compression ratio adjustment module, and decoder. The encoder includes an input layer, five one-dimensional convolutional (Conv1D) layers, and a MaxPooling layer. The compression ratio adjustment module includes a flattened layer, two fully connected layers (one of which adjusts the compression ratio), and a reshaped layer. The decoder includes an output layer, five one-dimensional convolutional layers, and an upsampling layer. The specific structure of the LCSAE model is shown in Figure 6.

The specific parameters of the model are shown in Table 1. After inputting 1000 × 1 LEMP waveform data through the input layer, the encoder part outputs feature data with a size of 469 × 8 after being processed by the first Conv1D layer (with 8 filters, and the size of each filter is 64), other layers, and so on. In the compression ratio adjustment module, the data of the previous layer is first flattened, and then the waveform features are subjected to variable compression mapping through the dense connected layer (The *x* in the output shape of the ninth layer in the table represents an adjustable variable, and the minimum compression feature can be set according to actual needs). Finally, the Reshape layer is used to transform the feature output shape into 8 × 16.

The 12th layer then begins to enter the decoder part, which uses one-dimensional convolutional and the upsampling layer to decode the compressed data. The structure of each layer corresponds to the encoder one by one, and finally reconstructs the LEMP waveform.

### 2.4. Model Hyperparameter

#### 2.4.1. Activation Function

The activation function is indispensable in the use of a neural network. A good activation function can greatly improve the learning ability and representation ability of neural network models. The commonly used activation functions are the Sigmoid, the hyperbolic tangent (*Tanh*), and the rectified linear activation unit (ReLU) [35]. Among them, the ReLU activation function has been widely used in deep neural networks due to its high computational efficiency, but when the input is negative, ReLU will face the problem of neuron death, resulting in the loss of some useful information, while *Tanh* can effectively avoid this issue. Consequently, we use the *Tanh* activation function in the convolutional layer of the LCSAE model, as shown in Formula (4):(4)$$\mathrm{Tanh}\left(x\right)=\frac{e^{x}-e^{-x}}{e^{x}+e^{-x}}$$

The output of the *Tanh* activation function is zero-centered, which avoids the problem of bias offset. For the LCSAE adopted in this study, good results can be achieved with this activation function.

#### 2.4.2. Training Loss Function and Optimizer

In the training of the neural network model, the loss function is used to measure the quality of the model training results. By comparing the difference between the network output after model training and the actual value, the training process of the model is guided, so that the network parameters are changed in the direction of lower loss values. The greater the gap between the predicted value of the network and the real value, the greater the loss value, and the loss value is zero when the two are completely consistent. This experiment is a regression task and we use mean squared error (*MSE*) as the loss function. Its calculation formula is as follows (5):(5)$$MSE=\frac{1}{n}\overset{n}{\underset{i=1}{\sum }}{\left(y_{i}-{\hat{y}}_{i}\right)}^{2}$$

At the same time, the update of network parameters is very important. The commonly used optimizers in deep learning are stochastic gradient descent SGD, momentum (Momentum), Nesterov accelerated gradient (NAG), RMSprop, adaptive moment estimation (Adam), and so on. Among them, Adam is equivalent to an optimization algorithm combining RMSprop and Momentum. It not only has a higher computing efficiency; but also has a small memory requirement. When faced with a problem with a larger amount of data, Adam has more advantages than other optimizers. Therefore, we use Adam to update the network parameters that affect the model training process and results, and the Adam learning rate is set to 0.001. Furthermore, for other parameters during network training, we set the batch size to 128 and the epochs to 100.

## 3. Results and Evaluation

This experiment uses the Keras deep learning tool to build the LCSAE model based on the Python3 language platform. A NVIDIA Quadro RTX 4000 GPU is used to increase the speed of the model to process data.

### 3.1. Evaluation Index

To conveniently demonstrate the compression effect of the model, we evaluated the experimental results using several evaluation metrics, including the compression ratio (*CR*), mean square error (*MSE*) (as defined by Equation (5)), coefficient of determination ($R^{2}$), and signal-to-noise ratio (*SNR*). Among them, *CR* represents the compression degree of the signal, as shown in Formula (6):(6)$$CR=\frac{l_{1}-l_{2}}{l_{1}}\times 100\%$$
where $l_{1}$ represents the original waveform length, and $l_{2}$ represents the compressed waveform length. The higher the *CR* value, the better the compression effect.

$R^{2}$ evaluates the fitting degree of the original waveform and the reconstructed waveform, as shown in Formula (7):(7)$$R^{2}=1-\underset{i}{\sum }\frac{{\left(y_{i}-{\hat{y}}_{i}\right)}^{2}}{{\left(y_{i}-{\bar{y}}_{i}\right)}^{2}}\;$$
where $y_{i}$ is the actual value, ${\bar{y}}_{i}$ is the average value of the actual value, and ${\hat{y}}_{i}$ is the predicted value.

The value range of $R^{2}$ is between 0 and 1. When *R* is closer to 1, it means that the fitting effect is better between the reconstructed waveform and the original waveform. Simultaneously, the model prediction effect is better. A waveform reconstructed by a good model method should be able to have a high $R^{2}$ at a high *CR*.

*SNR* evaluates the signal strength and the *i*-th signal is calculated by Formula (8):(8)$$SNR=\frac{\sum_{j=0}^{20}\left|x_{i\left(N-10+j\right)}\right|}{\sum_{j=1}^{1000}\left|x_{ij}\right|}$$
where *N* is the index value corresponding to the peak of the waveform and $x_{ij}$ is the *j*-th eigenvalue in the i-th waveform signal.

### 3.2. Experimental Results

The training set of LEMP waveform data is input into the LCSAE model for training, and the validation set is used to test the reconstruction results after each round of training. Then, the model parameters are updated according to the verification results to prepare for the next round of training, and ends after 100 rounds of training. The original length of the LEMP waveform is 1000, the compression range is between 1 and 128, and the corresponding *CR* is 99.9% to 87.2%. In the experiment, the LCSAE model is first used to compress and reconstruct the −CG. During the process of the experiment, the loss value *MSE* for training and validation are shown in Figure 7 and Figure 8. It can be seen that with the increase of epoch, the *MSE* generally shows a downward trend. At the same time, with the increase of *CR*, the corresponding *MSE* of each group also showed an increasing trend. This is because a larger *CR* corresponds to a smaller feature encoding of the model, and the corresponding compression difficulty will also increase.

After training, the effects of compression reconstruction of the model are tested using test set data. Figure 9 shows the comparison between the original waveform and the reconstructed waveform of the LEMP waveform data (the example in the figure is −CG) when the *CR* value is between 87.2% and 99.9%. It can be seen that the fit of the reconstructed waveform to the original waveform increases as the minimum compression feature increases. When the minimum compression feature is 1 (Figure 9a), the reconstructed waveform can express the main shape of the original waveform, and there are some differences in waveform peak location and amplitude. Compared with (a), the reconstructed waveform feature information of (b)~(e) shows a trend of increasing gradually; in particular, the reconstruction quality at the trough increases significantly. In Figure 9f–h, it can be seen that the fitting quality of the reconstructed waveform and the original waveform has reached a good level. Except for some environmental noise information, other features have no large deviation in timing and amplitude.

When the minimum feature is 64 (Figure 9g, *CR* = 93.6%), the average $R^{2}$ of −CG is 96.7%. In this case, the decoder part of LCSAE can reconstruct the waveform shape with very little difference from the original waveform through 64 minimum feature codes, and the *MSE* is 0.0317. In addition, when the minimum characteristic of LEMP signals is 64, the comparison between the original waveform and the reconstructed waveform is shown in Figure 10. It can be seen from the figure that the waveforms reconstructed using the compressed features have clear peaks, steep rising and falling edges, complete waveforms without missing segments, and can show different features of different types of waveforms. The ability of the LCSAE method to reconstruct different LEMP waveforms under such compression conditions can achieve good results.

The model was trained separately using the training sets of the six types of waveform data: −CG, CG-IR, +CG, −NBE, +NBE and SW, and the resulting *MSE* obtained in different experiments was between 0.0176 and 0.3210. As shown in Figure 11, it can be seen that the *MSE* shows an upward trend with the increase of *CR*. Meanwhile, it can be seen that the *MSE* rises slowly before *CR* = 93.6% (the corresponding compression feature is 64), but the rise of *MSE* increases significantly after *CR* = 93.6%. This is because fewer feature codes can express less information. With the reduction of the features extracted by the model, the key information of the LEMP waveform expressed by the minimum compression layer will also decrease, and the difficulty of waveform reconstruction will also increase. Furthermore, Figure 12 compares the experimentally obtained *MSE* and *SNR* results at *CR* = 93.6%. In the figure, *SNR*_O and *SNR*_R represent the average *SNR* of the original and reconstructed data, respectively. This shows that *SNR*_R is higher than *SNR*_O, with the smallest difference in SW and the largest difference in −NBE. The CG-IR category has both the smallest overall *SNR* and the largest *MSE* among the categories tested. Furthermore, the reconstruction quality of CG-IR signals is poorer than that of the other categories. This can be attributed to the higher complexity of CG-IR signals, which results in a lower *SNR* and makes feature compression more difficult.

Figure 13 is a comparison chart of the $R^{2}$ values of the six types of LEMP signals under different *CR*s. The $R^{2}$ of LEMP shown in Figure 13 can all reach more than 90% when *CR* = 93.6% (minimum feature = 64). At this time, the $R^{2}$ of the −CG signal is the largest at 97.78%, and the $R^{2}$ of CG-IR is the smallest at 94.25%. The average $R^{2}$ at this point reaches 96.7%. This indicates that when the minimum compression setting of the LCSAE model is set to 64, signals such as CG-IR experience more information loss during the encoding and compression process compared to other signals, resulting in a decrease in reconstruction accuracy. This is mostly caused by the CG-IR waveform, which has more pulses per event than other waveform types and is more complicated.

Summarizing the above discussions, comparing Figure 11 and Figure 13, it can be seen that the growth trend of $R^{2}$ and *MSE* is opposite, so the waveform fit decreases as the reconstruction error increases. When the compressed minimum feature is 64, the average coefficient of determination $R^{2}$ of the reconstructed waveform and the original waveform can reach 96.7%.

### 3.3. Test and Analysis

A good model should have the advantages of both experimental results and experimental time. Finally, the training process of the built LCSAE model is tested to analyze its performance for LEMP waveform data compression. The compression of the entire LEMP waveform data set was tested on a PC with a 64-bit operating system, Intel(R) Core (TM) i5-9400F CPU @ 2.90 GHz, and an NVIDIA Quadro RTX 4000 GPU. The single training time of each piece of data obtained from the test is about 37 μs. Meanwhile, we do a test on the real-time operation speed of the LCSAE model in the embedded platform. This device has mainly a Cortex-A53 processor and a memory connected to the processor. The memory stores the instructions to be executed, which are executed by the processor to implement the compression and reconstruction process of the LEMP waveform by using the new deep learning model in the paper. The LCSAE model that was trained on the PC was added to the processor of the embedded platform (In the ARM of Figure 3), and the average time to process the waveform was measured to be 198 ms.

Eventually, we conducted work to compare the peak locations of the original waveform with those of the reconstructed waveform for six types of LEMP data. As a result, Figure 14 illustrates the count of offsets for LEMP signals in the test set. As illustrated in Figure 14, 65.6% of the data had no offset, while 14.7% and 12.8% of the data had offsets with values of −1 and 1, respectively. The number of samples with offset values of −2 and 2 accounted for 1.8% each, and the remaining 3.3% had other offset values. Overall, 93.1% of the peak offsets were less than or equal to one point, indicating that the reconstructed data using LCSAE compression is suitable for subsequent lightning event analysis such as localization and classification.

## 4. Discussion and Conclusions

In order to solve the problem of compression and reconstruction of LEMP waveforms collected by lightning detection sensor network, this paper applies the deep learning method to LEMP data processing, and proposes an LEMP waveform compression and reconstruction model based on stacked autoencoders: LCSAE, which can compress the features of LEMP waveform data from the initial 1000 to 1~128, and the LEMP waveform can subsequently be reconstructed using the compressed feature code. The model uses the encoder module to remove redundant features and compress the waveform layer-by-layer; the decoder module decodes and reconstructs the LEMP waveform data from the least compressed features.

After testing the compression performance of the LCSAE model on LEMP waveforms with different compression ratios of *CR* between 87.2% and 99.9%, it is concluded that the training loss function *MSE* is between 0.0176 and 0.3210 on the +CG and −CG types of cloud-to-ground flashes, and the fitting degree $R^{2}$ of the original waveform and the reconstructed waveform is between 0.6969~0.9856. Regarding the two types of cloud flashes, −NBE and +NBE, the training loss function *MSE* is between 0.0166 and 0.2809, and the corresponding $R^{2}$ is between 0.7096 and 0.9843. At the same time, *CR* is negatively correlated with $R^{2}$, and the compression performance decreases with the increase of CR, which can also be seen in Figure 11 and Figure 13. Finally, when the minimum compression feature value is set to 64, the reconstructed waveform’s $R^{2}$ value reaches 96.7% compared to the original waveform. Furthermore, after testing with the validation set, we found that 93.1% of the results had a peak offset between the reconstructed waveform and the original waveform that was less than or equal to one point. This finding is promising for future research on the localization and analysis of lightning events using this technique.

In conclusion, the proposed deep learning model LCSAE has beneficial effects on the compression and reconstruction of the LEMP signal. Moving forward, we aim to incorporate additional data types and techniques into our research to further enhance the performance of the LCSAE model.

## Figures and Tables

**Figure 1 sensors-23-03908-f001:**
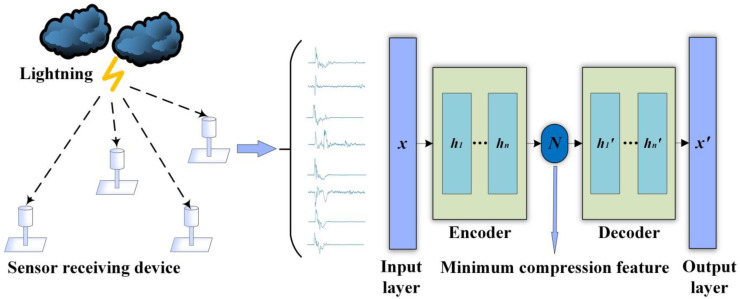
Schematic diagram of LCSAE waveform compression method.

**Figure 2 sensors-23-03908-f002:**
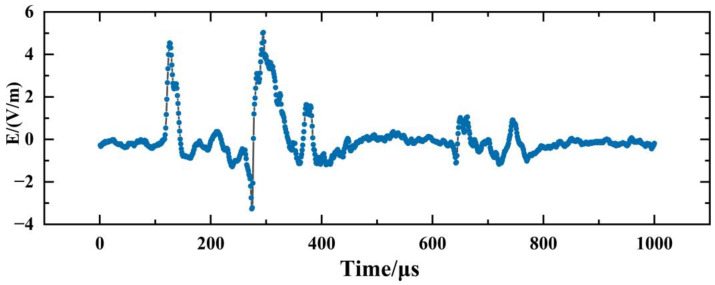
Schematic diagram of LEMP data collected by sensor detection equipment.

**Figure 3 sensors-23-03908-f003:**
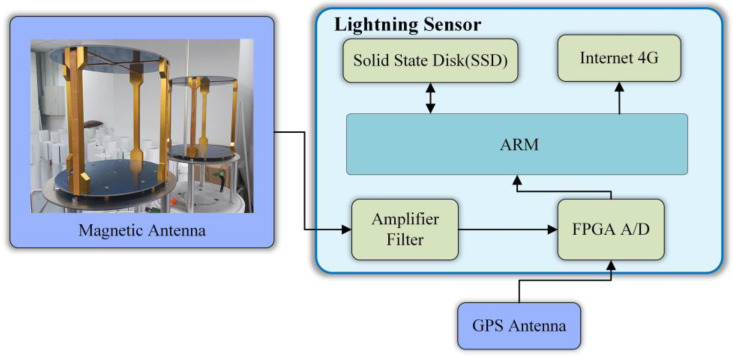
Block diagram of lightning sensor equipment components.

**Figure 4 sensors-23-03908-f004:**
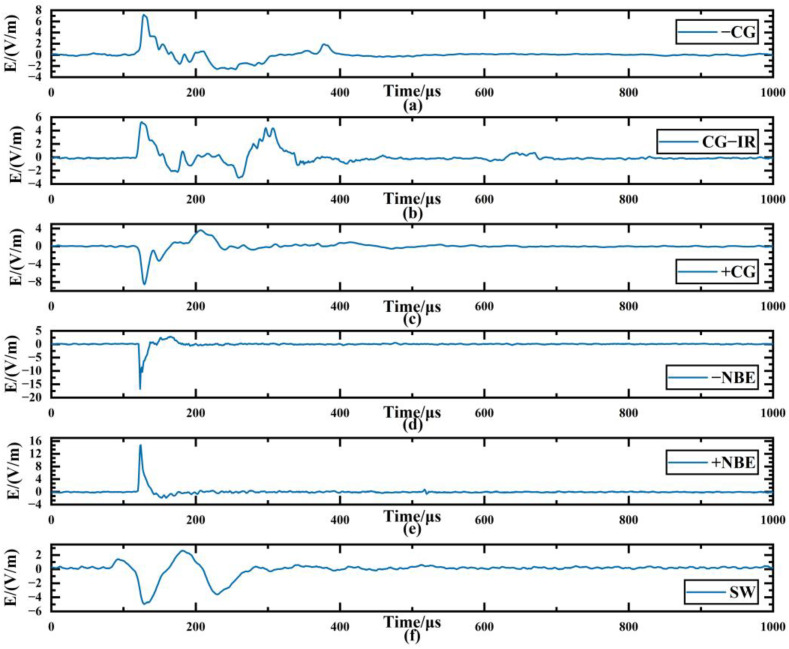
Examples of various lightning electromagnetic pulse waveforms. (**a**) negative cloud-to-ground flash (−CG); (**b**) positive cloud-to-ground flash (+CG); (**c**) negative narrow bipolar event (−NBE); (**d**) positive narrow bipolar event (+NBE); (**e**) cloud ground flash with ionosphere reflected signals (CG-IR); (**f**) far-field skywave (SW).

**Figure 5 sensors-23-03908-f005:**
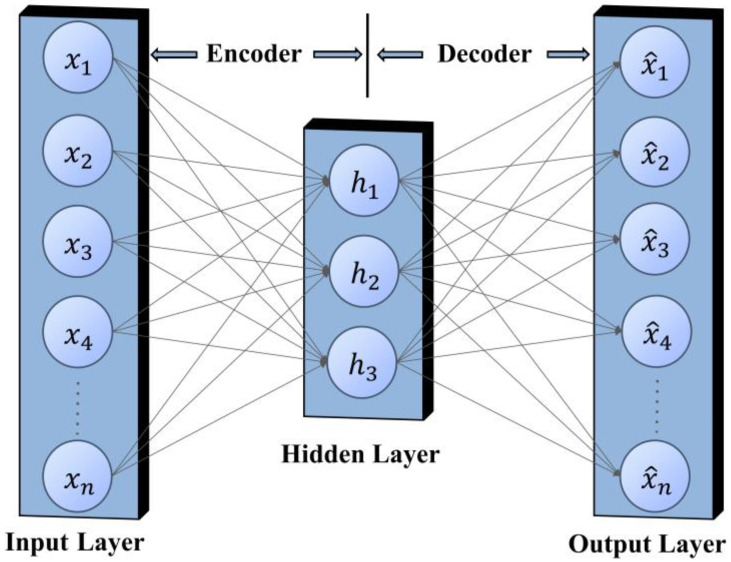
Autoencoder structure.

**Figure 6 sensors-23-03908-f006:**
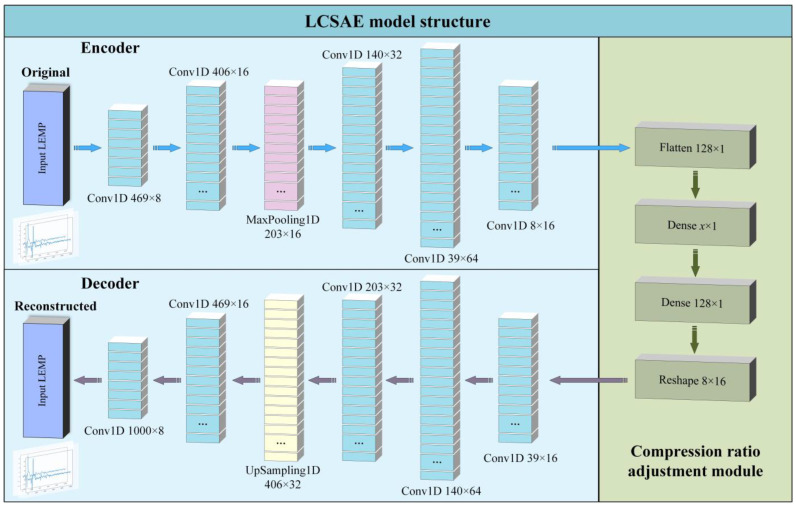
The block representation of the proposed LCSAE model for LEMP compression.

**Figure 7 sensors-23-03908-f007:**
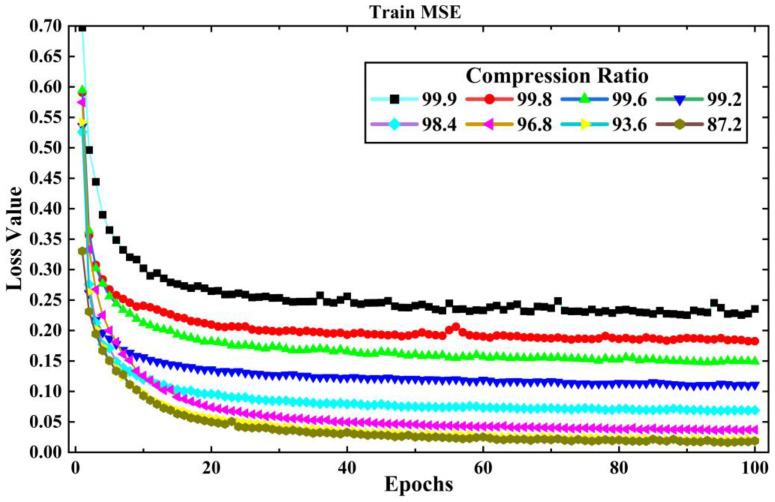
The changing of experimental loss value *MSE* with training epochs under different *CR* for a −CG lightning signal collected by the sensor array.

**Figure 8 sensors-23-03908-f008:**
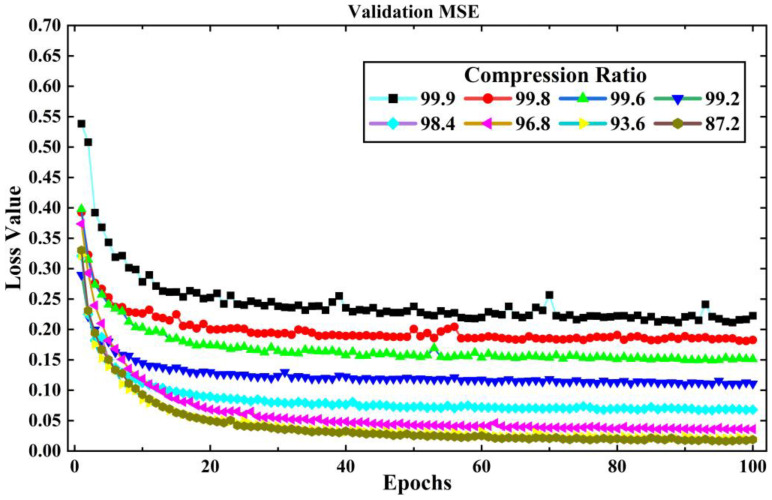
The changing trend of experimental loss value *MSE* with validation epochs under different *CR* for −CG lightning signal collected by sensor array.

**Figure 9 sensors-23-03908-f009:**
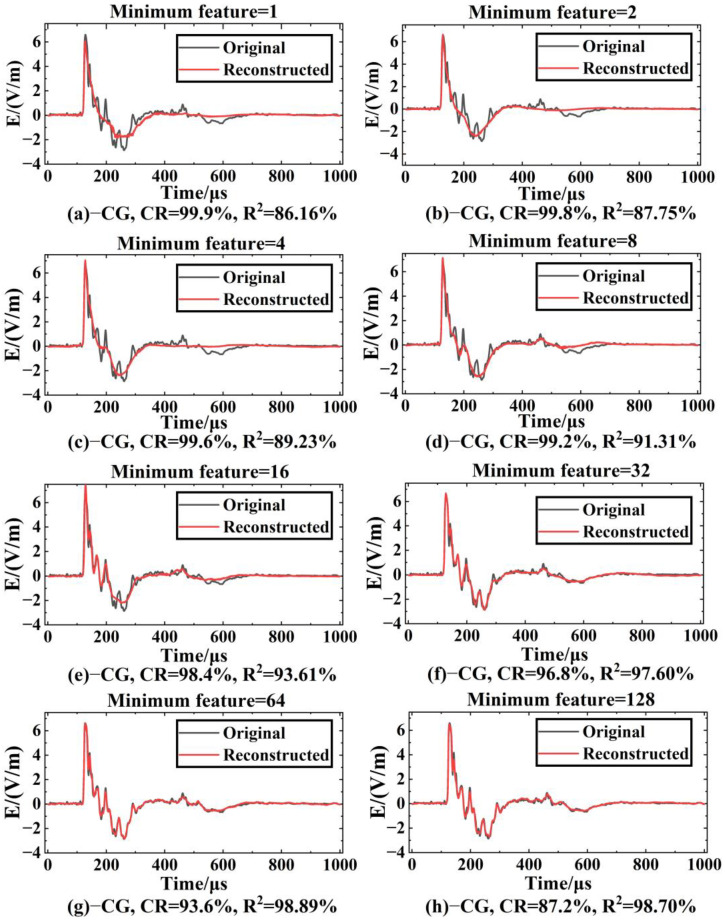
Comparison of original and reconstructed waveforms of −CG under different *CR*s.

**Figure 10 sensors-23-03908-f010:**
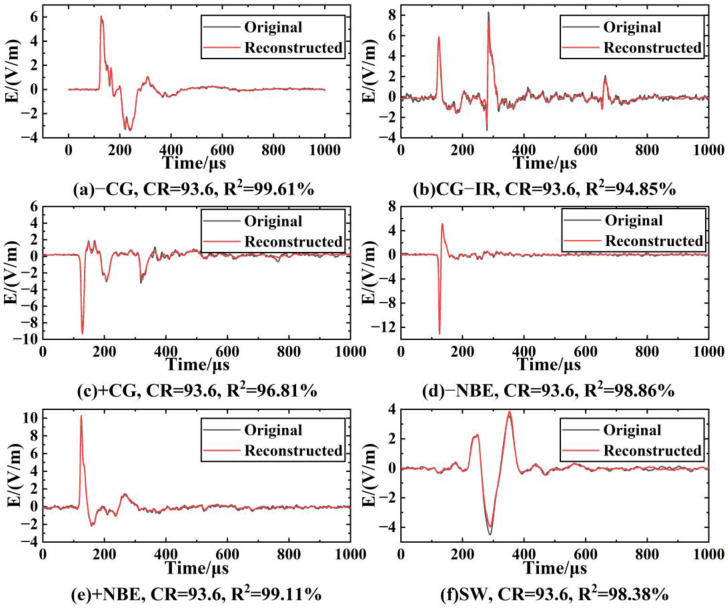
Comparison of original and reconstructed waveforms of LEMP at *CR* = 93.6%.

**Figure 11 sensors-23-03908-f011:**
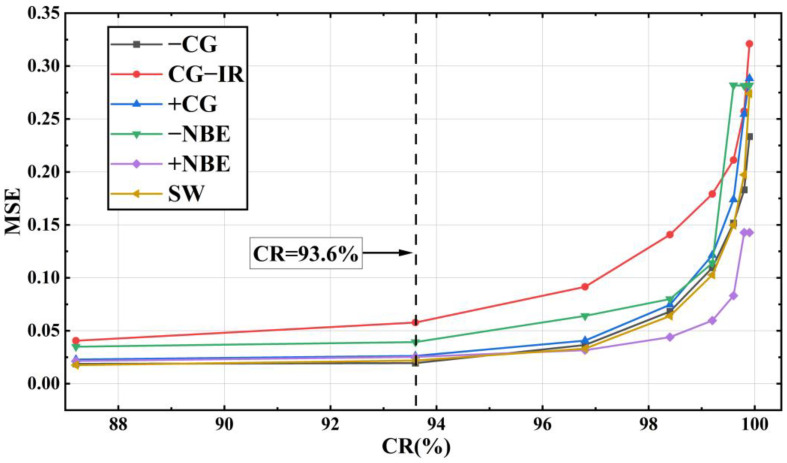
*MSE* comparison chart of LEMP waveforms after LCSAE model training.

**Figure 12 sensors-23-03908-f012:**
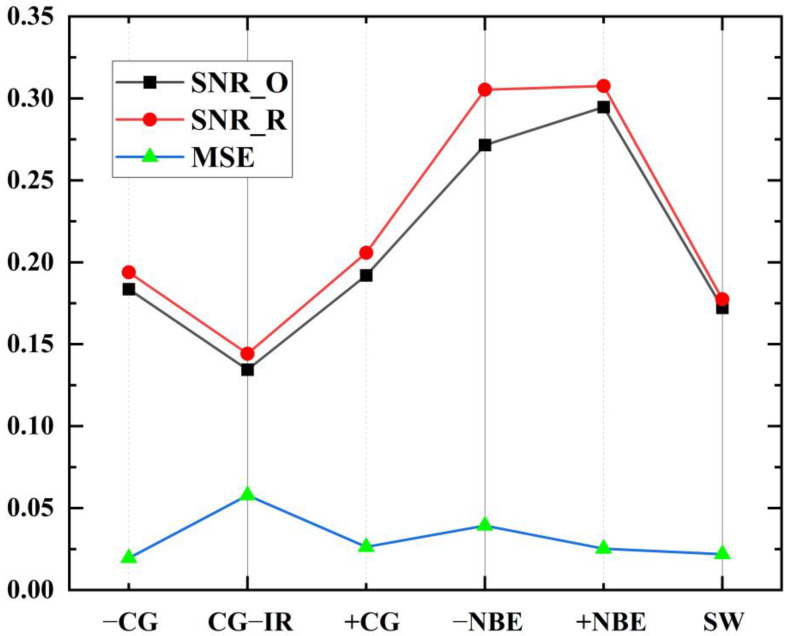
Comparison of the results of *MSE* and *SNR* obtained from the experiment at *CR* = 93.6%.

**Figure 13 sensors-23-03908-f013:**
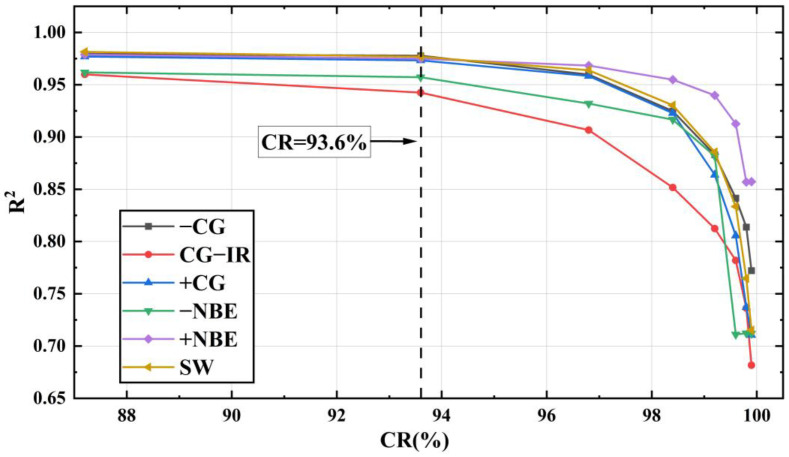
$R^{2}$ comparison chart of LEMP waveforms after LCSAE model training.

**Figure 14 sensors-23-03908-f014:**
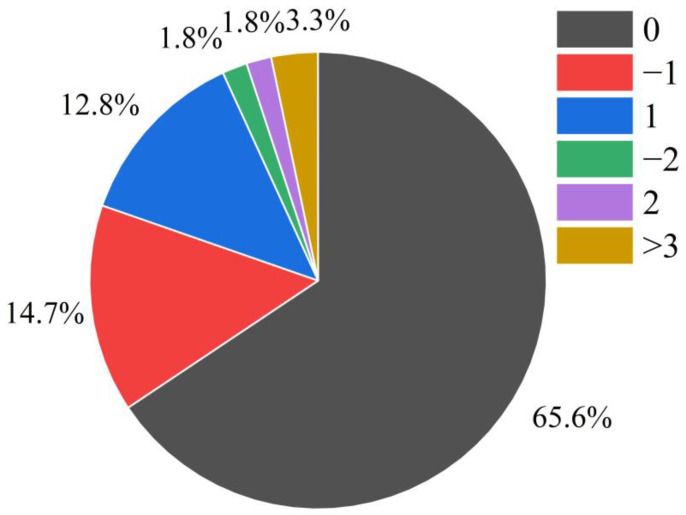
Statistics of the peak offset of the original LEMP waveform and the reconstructed waveform.

**Table 1 sensors-23-03908-t001:** Detailed parameters of LCSAE model structure.

Module	No.	Layer	Filter	Kernel Size	Stride	Activation Function	Output Shape
Encoder	1	Input	-	-	-	-	1000 × 1
	2	Conv1D	8	64	2	*Tanh*	469 × 8
	3	Conv1D	16	64	1	*Tanh*	406 × 16
	4	MaxPooling1D	-	-	-	-	203 × 16
	5	Conv1D	32	64	1	*Tanh*	140 × 32
	6	Conv1D	64	64	2	*Tanh*	39 × 64
	7	Conv1D	16	32	1	*Tanh*	8 × 16
Compression ratio adjustment module	8	Flatten	-	-	-	-	128 × 1
	9	Dense	-	-	-	*Tanh*	x × 1
	10	Dense	-	-	-	*Tanh*	128 × 1
	11	Reshape	-	-	-	-	8 × 16
Decoder	12	Conv1D	16	32	1	*Tanh*	39 × 16
	13	Conv1D	64	64	2	*Tanh*	140 × 64
	14	Conv1D	32	64	1	*Tanh*	203 × 32
	15	UpSampling1D	-	-	-	-	406 × 32
	16	Conv1D	16	64	1	*Tanh*	469 × 16
	17	Conv1D	8	64	2	*Tanh*	1000 × 8
	18	Output	1	1	-	Linear	1000 × 1

## Data Availability

Not applicable.

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
