# Peer review of "An Efficient Compression Method for Lightning Electromagnetic Pulse Signal Based on Convolutional Neural Network and Autoencoder"

_sensors, 2023, doi:10.3390/s23083908_

Round 1

Reviewer 1 Report

This paper presents the algorithm of compressing lightning waveforms using CNN and auto encoder. The paper fits the scope of the journal well and is worth being published. However, I have three major concerns and a list of editorial comments need to be addressed before recommending for publication.

Major concerns:

1.       The authors show the comparisons of waveforms before and after compression in figures and results look good. However, a very important point that needs to clarify is how the algorithm will distort the peak of the pulse. In other words, would the peak of the pulse shift after the compression-reconstruction process? Even 1-sample (1 microsecond for 1MS/s sampling rate) is not desired since such time shift will produce additional timing error and eventually lead to additional location error if you use time-of-arrival method for lightning locating. This need to be clarified.

2.       How do the results depend on the signal-to-noise ratio (SNR)? Authors did not state how strong the signals are for the waveforms in the dataset.  I believe the algorithm works well for strong pulses with good SNR. How about weak signals (which are more common in reality)? Does the performance downgrade? Such analysis is needed.

3.       Authors need to show how to apply this algorithm in practice. Will you run this algorithm on a computer attached to the sensor or on GPU or even FPGA? Is it fast enough to run in real time? Such information would be helpful.

Editorial:

L13: collected by VLF/LF instruments rather than electromagnetic pulse

A general comment, you do not need to give full name of first authors for each citation. Just last name is good enough.

L59: remove “that can be detected”

L61: major issue

L65: remove “in summary”

L65-79, this paragraph is a bit too long and conveys the same information as abstract. I recommend shortening it to be more concise.

L93, remove “receiving device”

L121, scrambled=>shuffled

L137, there is a format error “Error! Reference source not found” need to be fixed.

L207, question=>issue

A general comment, the quality of figures is too low. Need to replace them with high resolution ones.

Figure7, please also label CR AND R^2 in each subfigure

Reviewer 2 Report

The work is appreciable and would be useful to save time and storage space of the data. However, there is a concern about loosing the information while one transforms the electric field signals from time domain to frequency domain. This could be a crucial issue while analyzing the data in frequency domain and extracting the information in frequency domain. It is felt that further optimization of the computation is required.

Further, authors seem to be confused with the sensors and signals received by the sensors. These terms are mostly used inconsistently in the manuscript. It is advised that authors revise the text bearing in mind that you are applying the encoding and decoding technique on the lightning generated electric field and not on the sensors.

Please provide the information on the sensors that have been deployed for detection of lightning activity and of which waveforms were used in this study. 
